# Space-time smoothing of mortality estimates in children aged 5-14 in Sub-Saharan Africa

**Benjamin-Samuel Schlüter**[ID]*, **Bruno Masquelier**[ID]

Université Catholique de Louvain-la-Neuve (UCLouvain), Louvain-la-Neuve, Belgium

* benjamin-samuel.schluter@uclouvain.be

## Abstract

To meet the SDG requirement of spatial disaggregation of indicators, several methods have been developed to generate estimates of under-five mortality at the sub-national level. The reliability of sub-national mortality estimates in children aged 5-14 with the available survey data has not been evaluated so far. We generate Admin-1 sub-national estimates of the risk of dying in children aged less than five ($_5q_0$) and those aged 5 to 14 years old ($_{10}q_5$). We use 96 Demographic and Health Surveys (DHS) in 20 Sub-Saharan countries having at least 3 surveys designed to be representative at a sub-national level. The estimates account for the complex sample design of DHS and HIV-related biases in young children. A Bayesian space-time model previously developed for under-five mortality is used to smooth estimates across space and time in both age groups to reduce problems associated with data sparsity. The posterior distributions of the probability $_{10}q_5$ are used to compute coefficients of variation and assess precision. Sufficiently precise estimates are retained to study the sub-national relationship between age-specific mortality rates ($_5q_0$ and $_{10}q_5$), accounting for uncertainty in sub-national levels. Out of 1,132 space-time estimates, 62.3% are considered sufficiently precise with high heterogeneity across countries. Across all periods, sub-national estimates of mortality in children aged 0-4 are highly correlated with those in older children and young adolescents but this correlation is largely driven by the mortality decline. Within specific periods of time, it is often impossible to assess the relationship between mortality rates in the two age groups at the sub-national level, except in Nigeria, Ethiopia, Cameroon, Senegal and Zambia. As increased attention is devoted to survival after age 5, more research is needed to ensure that sub-national areas with specific interventions required for older children can be correctly identified.

## Introduction

Under-five mortality is one of the most important health indicators and is regularly targeted by global health policies. It has been the focus of intense research in low- and middle-income countries, with studies developing robust national and sub-national estimates from multiple sources [1–3], identifying the main determinants of child survival [4–6] and pinning down the most effective public health interventions to prevent child deaths [7, 8]. In comparison,

**Data Availability Statement:** The data are publicly available Demographic and Health Surveys that can be downloaded from https://www.dhsprogram.com/data/available-datasets.cfm.

**Funding:** This work was supported by a FSR research grant from Université Catholique de Louvain (UCLouvain): IACS FSR19 MASQUE, Mr. Benjamin-Samuel Schlüter.

**Competing interests:** The authors have declared that no competing interests exist.

mortality in older children and young adolescents (5 to 14 years old) has attracted less attention, presumably because risks of dying are much lower at these ages. According to the UN Inter Agency Group for Child Mortality Estimation (IGME), 5.2 million children under 5 years of age died in 2019, compared to 0.9 million children aged 5-14 [9]. Children and young adolescents aged 5-14 face the lowest risk of dying observed across the life course, estimated at 7 deaths per 1000 children in 2019 globally. Yet, from the perspective of child health and development, this age group is of particular importance as it marks the start of formal education and the onset of adolescence [10]. Deaths in children aged 5-14 also deserve more attention as they are predominantly caused by preventable conditions (including lower respiratory tract infections, diarrhoeal diseases, drowning, meningitis and road injuries) [11]. Since the 2000s, the decline in under-five mortality has greatly accelerated, while that in mortality in the age range 5-14 has not, raising concerns that older children might not be benefiting from recent efforts to improve child survival as much as neonates and children aged 1-59 months [12].

Sub-Saharan Africa has by far the highest rate of mortality in children aged 5-14 years, at 17 per 1000 children, but this regional average conceals wide variations across countries [9]. In Niger for example, the probability of a child dying between her fifth and fifteen birthday ($_{10}q_5$) was estimated at 30 per thousand in 2019, six times the risk in South Africa (at 5 per thousand). There could also be substantial heterogeneity at the sub-national level but to date no study has attempted to disaggregate mortality indicators for this age group. Within-country differences in mortality in younger children are striking, suggesting sub-national variation in 5-14 year old mortality too; they accounted for 74-78% of overall variation in under-5 mortality across space and over time in Sub-Saharan Africa in the 1980s, 1990s and 2000s [13]. Hence progress towards child survival goals needs to be tracked at the sub-national level, a need recognized in the Sustainable Development Goals (SDGs) resolution [14]. Sub-national estimates help inform effective resource allocation to reduce child health inequities within countries by targeting areas where interventions are most needed [15]. Health programme planning and implementation typically occur at the level of low administrative units, such as regions, provinces and districts [2].

In recent years, the need for sub-national estimates of under-five mortality has been met thanks to the increased sample sizes of Demographic and Health surveys [16] and the development of new statistical methods [2, 3, 17]. Subnationally, the number of deaths and the corresponding population at risk in any unit of space × time are often too small to produce reliable estimates of the underlying risk without "borrowing strength" from neighbouring units [18]. Different strategies have been employed by various research groups for under-five mortality. Most notably, Golding et al. [2] used a Bayesian geostatistical analytical framework to produce estimates of under-5 and neonatal mortality at a resolution of 5 × 5 km grid cells across 46 countries in Africa. The study team combined full birth histories (FBH) collected in large-scale household surveys with summary birth histories (SBH) collected in censuses, where women report only on the children ever born and surviving. They showed that many sub-national areas would not meet the SDG 3.2 target by 2030 as this would require reducing under-five mortality by at least 8.8% per year between 2015 and 2030, an unrealistic target. However, their approach had at least three limitations. First, their modelling strategy did not appropriately account for the complex survey design that is used in sample surveys, potentially introducing biases in the uncertainty around space-time estimates. Second, Golding et al. [2] did not correct for the potential underestimation of mortality in countries with large HIV epidemics due to mother-to-child transmission of the virus. As birth histories are collected from mothers, they do not represent the experience of children whose mothers have died. In the absence of treatment, children born to HIV-positive mothers face higher risks of death and are also less likely to appear in the sample. Third, while their continuous-spatial model permits

inference at various level of aggregation (districts, provinces, regions), it was less robust than other discrete spatial alternatives [3].

Another small area estimation (SAE) method was developed by Mercer and colleagues [17]. SAE methods leverage space-time similarity to construct a Bayesian smoothing model where spatial and temporal dependence are virtues that the estimation process can exploit [19, 20]. Mercer et al. [17] used hierarchical models performing spatio-temporal smoothing that fully accounted for the effects of the survey design on the uncertainty in place-time-specific estimates. In data-sparse situations, when one expects similarity (in space, time or both) this approach allows to disentangle signal from sampling variability. This can also be seen as a penalization, where large deviations from "neighbors" are discouraged. This method is of particular interest to study geographical heterogeneity in data-sparse contexts. It was first applied in a study of under-five mortality in regions of Tanzania [17] and has since been applied by Li et al. (2019) to generate estimates for Admin-1 sub-national areas in 35 countries in Africa [3].

In this article, we replicate the Li et al. (2019) study but focus on mortality in older children and adolescents. Spatial variations in children aged 5-14 might not be strongly associated with those in children younger than 5 years, due to different cause-of-death patterns or risk factors. There might be a need to target areas with specific interventions for older children to reach those most at need in this age range. However, data sparsity is a more serious problem for older children, as fewer data sources provide mortality measures for this age group. For example, indirect demographic techniques cannot be used to estimate the probability $_{10}q_5$ from summary birth histories collected in censuses. In addition, mortality rates are much lower in older children and confidence intervals are larger. For example, in the 2018 DHS conducted in Zambia, the under-five mortality rate for the period 0-4 years before the survey was 58.1 per thousand at the national level, with a standard error of 3.6 per thousand (coefficient of variation = 6.3%). The corresponding risk of dying in the age group 5-14 was 10.0 per thousand, with a standard error of 1.4 (CV = 14.0%). For this recent period, women aged 15-49 in the survey reported more than 7 times more deaths among children aged 0-4 years than among children aged 5-14 years, for a relatively comparable exposure time (1.5 times more person-years in children aged 5-14). In this context, our study addresses two questions. First, we assess whether the method developed by Mercer et al. [17] can yield sufficiently precise mortality estimates of $_{10}q_5$ at the first sub-national level when applied to Demographic and Health surveys (DHS). Second, for countries and periods for which we obtain estimates with enough precision, we evaluate if there is a strong relationship between under-five mortality and mortality in children aged 5-14 at the first sub-national level.

## Materials and methods

### Data sources

This study is based solely on DHS. DHS surveys use a stratified two-stage cluster sampling design. Samples are typically stratified by regions and urban/rural areas within each region. In each stratum, enumeration areas (EAs) are selected with a probability proportional to their population size, using a sampling frame usually obtained from the most recent census. Households are selected in each of the clusters. Within each household, all women aged 15-49 are selected for in-depth individual interviews. In recent surveys, geospatial information is collected to identify the central point of each cluster. To ensure that respondent confidentiality is maintained, coordinates are displaced by up to 2 km in urban clusters, 5 km in 99% of rural clusters, and 10 km in a random sample of 1% of rural clusters [21].

For this study, we retained only surveys conducted in Sub-Saharan countries that had at least 3 DHS surveys representative at a sub-national level, with a consistent set of sub-national boundaries across DHS surveys. This requirement was necessary to pool information from different surveys across time. When sub-national boundaries changed across DHS surveys, we used the GPS location of clusters to re-locate them into coherent sub-national units over time. This led us to select 96 surveys (see S1 Table in S1 Appendix) conducted in 20 Sub-Saharan African countries between 1990 and 2018. Our study covers the following countries: Benin, Burkina Faso, Cameroon, Ethiopia, Ghana, Guinea, Kenya, Lesotho, Madagascar, Malawi, Mali, Namibia, Niger, Nigeria, Rwanda, Senegal, Tanzania, Uganda, Zambia and Zimbabwe. The population-weighted average of the risk of dying in the age group 5-14 was 20 per 1000 in 2019 in these 20 countries, which is slightly higher than the regional estimate for Sub-Saharan Africa (16 per 1000) [9]. In 2019, 65% of all deaths that occurred in Sub-Saharan Africa in the age group 5-14 where in these 20 countries. The selected surveys contained data on about 11 million child-years of exposure and 376,015 deaths of children under the age of 5 years, against 9.2 million child-years of exposure and 20,605 deaths in the age group 5-14.

## Estimation of sub-national mortality rates

We used the complete birth histories of women included in the DHS to obtain child-month data where the event of interest was the death of the child. We followed the methodology developed in Mercer et al. (2017) [17] and Li et al. (2019) [3] to estimate sub-national-period specific mortality rates, both for children aged 5 to 14 and for those aged less than five. The methodology consists of two steps. First, we obtained place-time probabilities of dying for both age groups. Then, we used these as inputs in a space-time Bayesian model to smooth estimates across space and time.

In the first step, we used discrete time survival analysis (DTSA) [22] to estimate age-specific monthly probabilities of dying using weighted logistic regression [23]. For under-five mortality, we used 6 age bands: [0, 1) to identify neonatal deaths, [1, 12), [12, 24), [24, 36), [36, 48) and [48, 60). For children aged 5-14, we used 10 one-year age groups: [60, 72), [72, 84), . . . until [168, 180) months. The DHS sample weights associated with each mother account for the sampling probability and a non-response adjustment. Hence, the survey design and the related uncertainty is propagated through to the estimates and their associated standard error. This resulted in survey-place-time-age-specific estimated monthly probabilities of dying with variance estimates that fully account for the design of the survey. In our context, place consisted of Admin 1 level, which corresponds to administrative boundaries at the first sub-national level. DHS surveys are powered such that most indicators can be considered representative at this aggregation level. The time dimension of our mortality estimates was defined as a series of 4-year periods, for which multiple surveys could provide information. Because we need to combine surveys whose fieldwork dates have varied over time, we decided to let the most recent survey defines the time breaks (see S1 Appendix). For each survey, we went backwards in time by periods of 4 years. Information contained in full birth histories and referring to a period located more than 24 years before data collection was discarded when estimating under-five mortality. Similarly, this period was restricted to a maximum of 12 years for mortality in children aged 5-14 to reduce biases associated with the excess mortality of first-born children [12]. In other words, a survey contributed to a maximum of 6 periods for $_5q_0$ and 3 periods for $_{10}q_5$. Each country has its own series of 4-year periods. Note that if a survey provided only 1 year of exposure in a given 4- year period, it did not inform estimates in this period. We constructed the estimate of interest $_{10}\hat{q}_5^{its}$, in Admin 1 $i$, time period $t$ and survey $s$

using standard life table techniques:

$$
\begin{aligned}
{}_{10}\hat{q}_5^{its} = \ & 1 - \prod_{x=60}^{179}(1 - {}_1\hat{q}_x^{its}) \\
= \ & 1 - \{[\frac{1}{1 + exp(\hat{\beta}_1^{its})}]^{12} \times [\frac{1}{1 + exp(\hat{\beta}_2^{its})}]^{12} \times \ldots \times [\frac{1}{1 + exp(\hat{\beta}_{10}^{its})}]^{12}\}
\end{aligned}
\tag{1}
$$

The $\beta$'s are obtained from the weighted logistic regressions as follows:

$$
log(\frac{{}_1q_x^{its}}{1 - {}_1q_x^{its}}) = \beta_a^{its}
\tag{2}
$$

After estimation of parameters in Eq (2), it can be rewritten as

$$
{}_1\hat{q}_x^{its} = \frac{exp(\hat{\beta}_a^{its})}{1 + exp(\hat{\beta}_a^{its})}
\tag{3}
$$

for $x \in [x_a, x_a + n_a]$, $a = 1, \ldots, 10$, which consists of the 10 age intervals. Variances of the ${}_{10}\hat{q}_5^{its}$ were obtained via the delta method approach, using the estimated covariance matrices available from the weighted logistic regression models [17]. Under-five mortality rates were estimated in a similar way.

Each of our selected countries had a least 3 DHS, thus for a given period, and a given Admin 1, ${}_{10}\hat{q}_5^{it}$ and ${}_5\hat{q}_0^{it}$ were potentially derived from multiple surveys. We combined estimates from different surveys into a single estimate for each age group using weighted average, where weights were given by the inverse of their variances. Resulting estimates can be seen as standard meta-analysis estimates. Similarly to Li et al. (2019) [3], we adjusted the estimates of ${}_5\hat{q}_0^{it}$ to account for downward biases introduced by mother-to-child transmission of HIV in Kenya, Lesotho, Tanzania, Zambia, Zimbabwe, Malawi, Namibia and Rwanda [24]. We used the correction factors made publicly available by Li and colleagues [3]. We did not correct for HIV-related biases in ${}_{10}\hat{q}_5^{it}$ as these biases has not been evaluated in older children and there is currently no method to adjust for these.

In a second step, we modeled the entire collection of place-time specific logit($_5\hat{q}_0^{it}$) and logit $(_{10}\hat{q}_5^{it})$ for each country, using a Bayesian space-time smoothing model with appropiate design-based estimator of the variance, $\hat{V}_{it}$, obtained in the previous step. The smoothing allows sharing information between near neighbors in both space and time, leading to estimates that are characterized by a smaller variance and higher signal to noise ratio [3]. The model can be expressed as follows:

$$
\begin{aligned}
logit(_{10}\hat{q}_5^{it}) \ & \sim \mathcal{N}(\lambda_{it}, \hat{V}_{it}) \\
\lambda_{it} \ & = \mu + \alpha_t + \gamma_t + \theta_i + \phi_i + \delta_{it}
\end{aligned}
\tag{4}
$$

where the components are:

- A constant overall level $\mu$ with a flat prior.

- An unstructured spatial term with prior $\theta_i \sim_{iid} \mathcal{N}(0, \sigma_\theta^2)$, $i = 1, \ldots, n$.

- A smooth spatial term $[\phi_1, \ldots, \phi_n]$ where its prior is expressed as an intrinsic conditional autoregression (ICAR) model [25]. This model is a generalization of the random walk of order 1 (RW1) to the space dimension.

- An unstructured temporal term where the prior is expressed as $\alpha_t \sim_{iid} \mathcal{N}(0, \sigma_\alpha^2)$, $t = 1, \ldots, T$.

- A smooth temporal term $[\gamma_1, \ldots, \gamma_T]$ where its prior is a random walk of order 2. A random walk of order 2 (RW2) model generally leads to more smoothing than a random walk of order 1 (RW1) model. Indeed, in the later, each value depends on its nearest neighbors while in the former, it also depends on neighbors' nearest neighbors.

- $\delta_{it}$, an interaction term (type IV) where we assumed that temporal trends differ between areas and that they are more likely to be similar for adjacent areas [18]. Its associated prior is expressed as the product of the random walk and the intrinsic conditional autoregression.

Hyperpriors on the dipersion parameters have Gamma(a,b) distribution. For a detailed explanation of how a and b are specified, we refer the reader to [17, 18]. As emphasized by Li et al. (2019) [3], for both space and time dimensions, there are two terms capturing different sources of variation. The first term refers to unobserved risk factors that vary smoothly ($\phi_i$ and $\gamma_t$), the other reflects random shocks that are specific to that place or time only ($\theta_i$ and $\alpha_t$). The Bayesian estimation process takes place separately for each country. The amount of smoothing is determined by the variances of $\phi_i$ and $\gamma_t$ that are directly estimated from the data. The space-time interaction term ($\delta_{it}$) allows the trend in time to vary across spatial areas. It was modeled assuming that the temporal (RW2) and spatial (ICAR) structured effects interact [18]. In summary, each $logit(_{10}\hat{q}_5^{it})$ (or $logit(_5\hat{q}_0^{it})$) depends on its nearest neighbors in space and its nearest two neighbors in both the past and future in the time dimension.

In the S1 Appendix, we compare the estimates of $_{10}q_5$ and their associated precision before and after space-time smoothing (that is, after the first and second step). This comparison gives a sense of the reduction in uncertainty induced by the smoothing model. We also compare our final estimates at the national level with those provided by the UN Inter-agency Group for Child Mortality Estimation (UN IGME), using other data series and methods [9].

## Precision of sub-national estimates of $_{10}\hat{q}_5$

Our first objective is to assess if Admin 1 estimates are sufficiently precise for children aged 5-14 when derived from multiple DHS. In the frequentist framework, the coefficient of variation (i.e. the standard deviation divided by the mean) is regularly employed to decide on the usability of estimates [16]. Hansen et al. (1953) and Kish (1995) consider that a coefficient of variation larger than 0.2 is a sign of unstable estimate [26, 27]. Working in a Bayesian framework, we used the posterior distributions of sub-national mortality estimates and computed a coefficient of variation by dividing their standard deviations by their means. We multiplied these ratios by 100 to obtain percentages. Dong and Wakefield (2004) showed that there is a direct relationship between the posterior CV obtained in this way and the ratio of higher end to lower end of the posterior credible interval [28]. If the coefficient of variation was higher or equal than 20%, we considered that the associated sub-national mortality estimate for children aged 5-14 was not sufficiently precise.

## Correlation between $_{10}\hat{q}_5$ and $_5\hat{q}_0$

Our second objective is to examine the relationship between $_{10}\hat{q}_5$ and $_5\hat{q}_0$ at the sub-national level. To do so, we retained only the periods for which at least three quarters of the sub-national mortality estimates of $_{10}\hat{q}_5$ were considered as sufficiently precise according to the CV. This threshold was set according to the countries with fewer Admin 1 entities. In the case of Malawi (consisting of three Admin 1 entities), applying a 75% rule, a period could not be retained if one sub-national estimate within this period was not robust. For Rwanda (characterized by five Admin 1 entities), applying the same rule, if more than one sub-national estimate in a given period was not robust, this period was not retained. Having our set of selected

periods for each country, we first computed Pearson correlation coefficients between $_{10}\hat{q}_5$ and $_5\hat{q}_0$ by country. We further stratified by period. Working in a Bayesian inference setup, $_{10}\hat{q}_5^{it}$ and $_5\hat{q}_0^{it}$ were associated to their respective posterior distributions. This allowed us to obtain 95% credible intervals around Pearson correlation coefficients, accounting for the uncertainty in our sub-national estimates.

Analyses were run with the R software version 3.6.1. We used the *SUMMER* package [29] that uses the *svyglm* function from the *survey* package [30] and the *R-INLA* package [31] to fit the models. All the R codes used to perform the analysis is available on Github: https://github.com/benjisamschlu/Subnational-5-14-MR.

# Results

## Sub-national mortality estimates for children aged 0-4 and 5-14 years old

The estimates presented in this section are obtained after space-time smoothing of meta-analysis estimates. Figs 1 and 2 present Admin 1 mortality rates and their associated 95% credible intervals for Nigeria and Kenya. The corresponding graphs for other countries are the S1 Appendix. For most countries and Admin 1 units, mortality rates declined in young children aged 0-4 over the periods analysed. There were fewer cases with a significant decline over the period of observation in older children and young adolescents (40.2% of admin-1 units over the 20 countries against 81.7% in under-five mortality, when comparing the first and last periods). However, the low percentage in 10q5 is also a consequence of the wide confidence intervals for the first and last periods. The sub-national patterns observed in under-five mortality cannot be readily generalised to older children. There is substantial heterogeneity in the $_{10}q_5$-to-$_5q_0$ relationship.

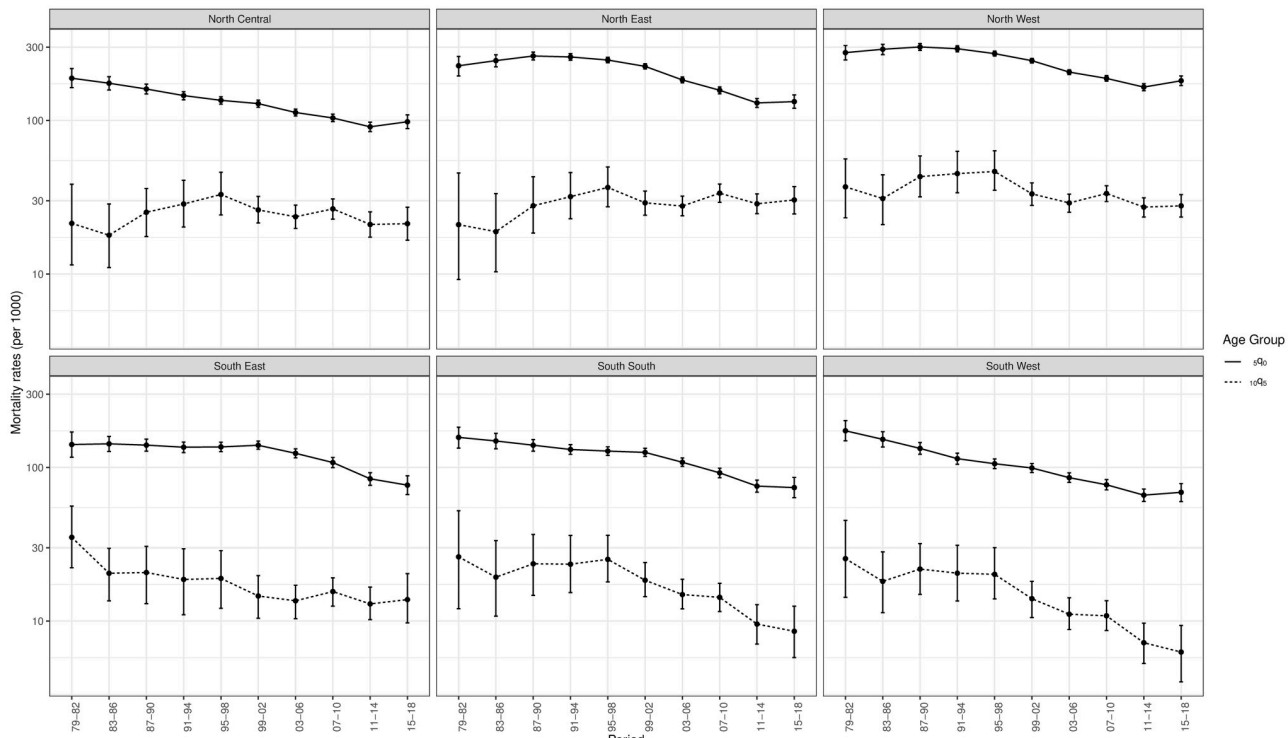

**Fig 1. Sub-national estimates of the probabilities $_5q_0$ and $_{10}q_5$ in Nigeria, based on DHS conducted in 1990, 2003, 2008, 2013 and 2018 (log scale).**

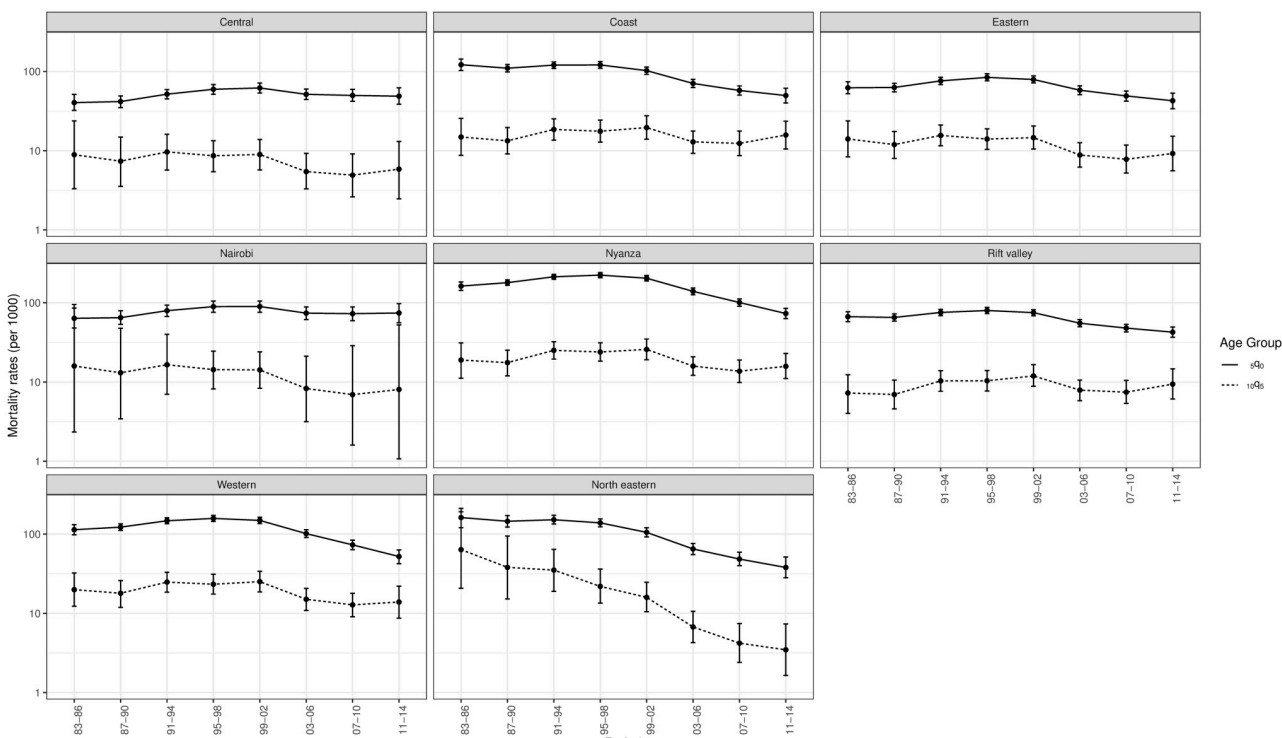

**Fig 2. Sub-national estimates of the probabilities $_5q_0$ and $_{10}q_5$ in Kenya, based on DHS conducted in 1993, 1998, 2003, 2008 and 2014 (log scale).**

The uncertainty around sub-national mortality rates for children aged 5-14 is much higher than for children aged less than 5. First, deaths counts are subject to more stochastic variation as the mortality rates for older children are smaller. Second, in the older age group, mortality estimates associated to a period are generally informed by less DHS surveys. This comes from the fact that 3 and 6 periods in the past are retained from each DHS in the case of older and younger children, respectively. Uncertainty is particularly large for the first and last periods. This is because these estimates are built on fewer DHS surveys and oldest surveys are characterized by smaller sample sizes. When the Admin 1 refers to a capital city, estimates tend to be associated with large credible intervals due to the smaller sample size and smaller count of deaths in this area. This can be observed for Nairobi in Fig 2.

### Precision of sub-national mortality estimates of children aged 5-14

Fig 3 displays the coefficient of variation for the probability $_{10}\hat{q}_5$ estimated for each Admin 1-period combination. These CVs are compared to the threshold value of 20%. All countries exhibit a U-shape, as sub-national mortality estimates for the first and last periods are less precise than the ones in the middle. Out of the 1,132 space-time estimates of $_{10}\hat{q}_5$, 62.3% fall below the 20% threshold. By contrast, this is the case of 99.3% space-time estimates of under-five mortality (see S1 Appendix). There is important heterogeneity in the amount of sub-national estimates that fulfill our criterion. In Zimbabwe, Kenya, Ghana, Lesotho and Namibia, sub-national estimates of the probability $_{10}\hat{q}_5$ are not sufficiently precise as most values of the CV fall above the 20% threshold. These countries had generally small sample sizes and a large number of Admin 1 entities which increases the amount of noise in the data.

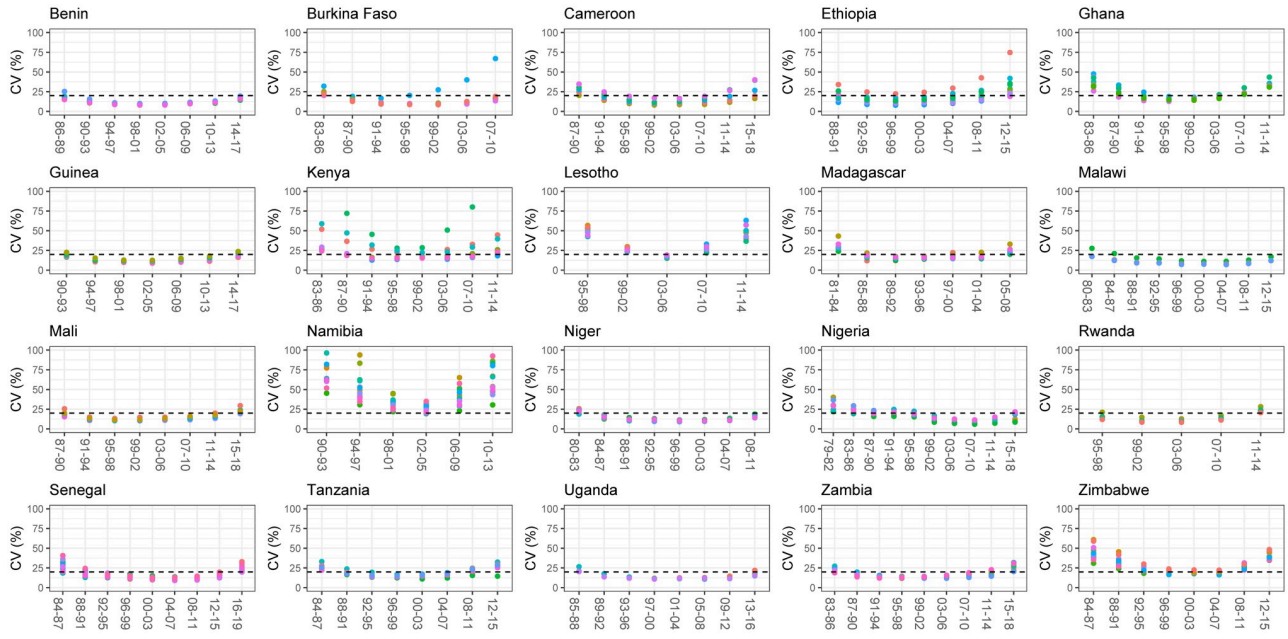

**Fig 3. Coefficient of variation associated with sub-national $_{10}q_5$ estimates.**

Fig 4 highlights the uncertainty in the sub-national estimates of $_{10}\hat{q}_5$ for two countries. In Ethiopia (top panel), a majority of coefficient of variations are above 20% for the period 2008-2011. This is due to wide posterior distributions for the sub-national mortality estimates, making it impossible to detect variations in mortality in older children within the country. On the contrary, sub-national estimates for Nigeria in the period 2007-2010 are all considered robust, which is due to much narrower posterior distributions, allowing detection of significant variations in $_{10}\hat{q}_5$. This is less so for the period 2015-2018 as it consists of the last period, and hence, estimates are built on fewer data points.

In order to study the relationship between $_5\hat{q}_0$ and $_{10}\hat{q}_5$, we retained periods where at least 75% of the $_{10}\hat{q}_5$ had a coefficient of variation below 20% for a given period. We also checked how many periods were considered sufficiently precise when using a threshold of 15% (see S61 Fig in S1 Appendix).

## The $_{10}q_5$-to-$_5q_0$ relationship at the sub-national level

Fig 5 displays the $_5\hat{q}_0$ estimates against the $_{10}\hat{q}_5$ values, each with their associated 95% credible intervals (note that the scale is defined specifically for each country and that different point shapes for a country corresponds to different periods). This Figure is based only on the periods for which we have sufficient precision in estimates of $_{10}\hat{q}_5$ according to our criterion. Except Lesotho and Zimbabwe, each other retained country has at least 4 periods considered as sufficiently precise. The figure allows to compare uncertainty in sub-national $_{10}\hat{q}_5$ estimates across countries. It shows the long term improvement in mortality for both age groups over time.

Pooling all sub-national estimates across retained periods, there is a clear positive and statistically significant Pearson correlation for all countries except Burkina Faso and Niger (see Fig 6). The correlation coefficient ranges from 0.14 in Niger (95% CI: -0.02—0.31) to 0.92 in Malawi (95% CI: 0.86—0.96). However, One should not conclude from these high coefficients that the spatial variation in mortality in older children is well reflected by that in under-five

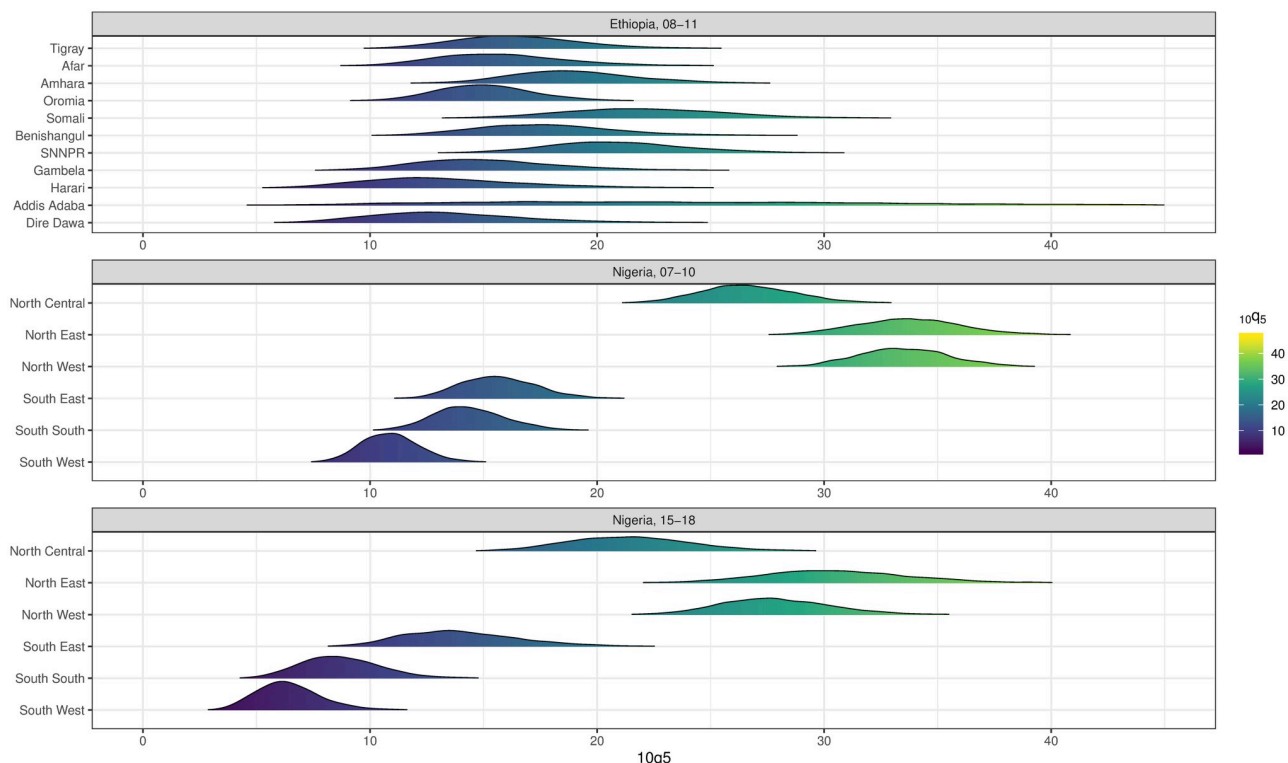

**Fig 4. Ridge plots of sub-national estimates of $_{10}\hat{q}_5$ for Ethiopia (2008-11) and Nigeria (2007-10, 2015-2018).**

mortality. Indeed, the relationship captured here is driven by the long-term improvement in mortality in both age groups. Stratifying by period, this relationship is much less obvious. Out of 82 retained periods, 46 have a Pearson correlation where its 95% credible interval crosses zero. In other words, in the majority of periods, the Admin 1 characterized by the highest under-five mortality rate are not the ones facing the highest mortality rate in older children. We can differentiate three groups of country. The first group consists of Cameroon, Ethiopia, Nigeria, Senegal and Zambia where correlations are most of the time positive and significantly different from zero. The second group contains Madagascar, Mali, Niger, Rwanda, Tanzania and Uganda when occasionally, a period is characterized by a positive and significant correlation. Finally, in the last group encompassing Benin, Burkina Faso, Ghana and Guinea, we did not find any significant correlation. We did not compute the Pearson correlation estimates stratifying by period for Malawi as the country consists of only three Admin 1 entities. Zimbabwe and Lesotho are not plotted as they only have one period considered sufficiently precise. Correlations associated with these periods are not significantly different from zero.

## Comparison of space and time heterogeneity among age groups

The total variance in $logit(_{10}\hat{q}_5)$ (or $logit(_5\hat{q}_0)$) can be split into the variance of the five components of the model (see Eq (4)). Pooling countries together, the variance associated to the RW2 time trend ($\sigma_\gamma^2$) makes up for an average of 58.86% and 46.44% of the total variance for $_5\hat{q}_0$ and $_{10}\hat{q}_5$, respectively (see Table 1). Countries with specifically high values for $\sigma_\gamma^2$ such as Benin, Rwanda, Ethiopia and Malawi, are the ones where all Admin 1 experienced a similar temporal trend, explaining the majority of the variation in sub-national estimates. Lesotho, Namibia

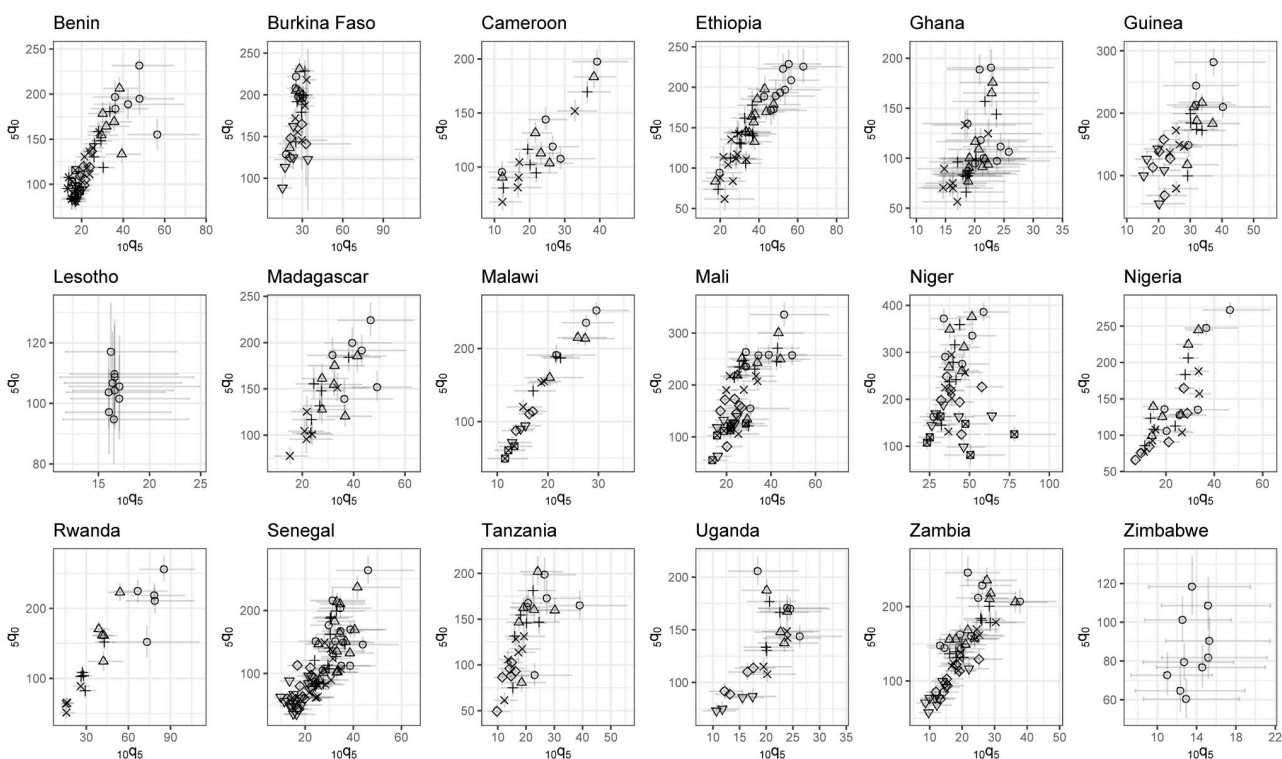

**Fig 5. The $_{10}q_5$-to-$_5q_0$ relationship at the sub-national level, based on space-time estimates considered sufficiently precise (scales varying by country).**

and Niger are countries where $\sigma_\gamma^2$ for $_{10}\hat{q}_5$ is below 10% which, in the case of Lesotho and Niger, is in contrast with what is found for $_5\hat{q}_0$.

The average share of total variance accounted for by the ICAR structured space term ($\sigma_\phi^2$) is 26.9% and 12.54% for $_5\hat{q}_0$ and $_{10}\hat{q}_5$, respectively. In all countries but Madagascar, Malawi, Uganda and Zambia, our estimates of $\sigma_\phi^2$ for children under the age of five are higher or equal than for older children, sometimes by a large margin (e.g. Ghana, Guinea, Kenya, Niger and Zimbabwe). This is a sign that sub-national patterns of mortality declines that are consistent over time account for less variance in sub-national mortality estimates for children aged 5 to 14 in comparison to children aged $< 5$. It seems that most of the "lost" variation from this component in the older age group is now accounted for by the time×space interaction (RW2xICAR). The average percentage of variability captured by $\sigma_\delta^2$ equals 8.48% in children under the age of 5 and 33.40% in older children and young adolescents. This suggests that particular Admin 1 within a given country had mortality estimates that deviated from levels expected based on the secular trend (applying to all Admin 1) and the sub-national pattern (applying to all periods). However, we have to be cautious as most countries with the highest values in $\sigma_\delta^2$ are also the ones that had few precise sub-national estimates (Burkina Faso, Ghana, Lesotho, Namibia and Zimbabwe). Hence, this might also comes from stronger noise in sub-national estimates for theses countries. Finally, the variability explained by the unstructured terms ($\sigma_\alpha^2$, $\sigma_\theta^2$) is within 1-5%.

## Discussion

The monitoring of progress in child survival is increasingly moving from the national to the sub-national level, or even to finer spatial resolutions. At the same time, increasing attention is

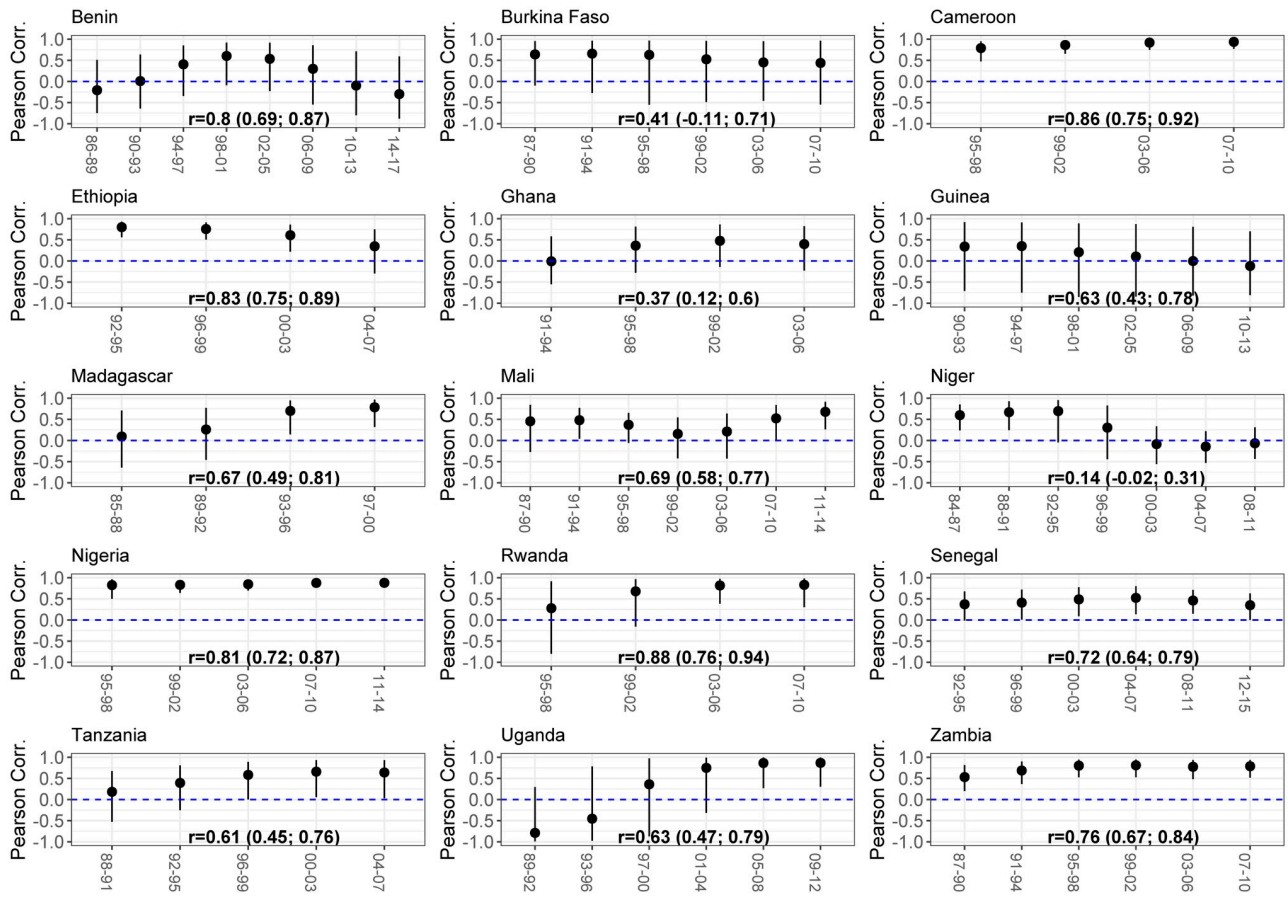

**Fig 6. Pearson correlations between Admin-1 estimates of $_5q_0$ and $_{10}q_5$.**

being paid to survival beyond the fifth birthday to ensure that older children can benefit from health programs and policies as much as young children [32]. In this context, it is important to assess whether spatial disparities in risks of dying in the early years of life mirror those that characterize mortality in older children. Otherwise, the targeting of public health interventions based solely on under-five mortality might leave behind some older children and young adolescents in need.

In this study, we showed that sub-national mortality estimates can be derived from DHS surveys for children aged 5-14. We used a method which has been validated for children aged 0-4 years [17], and obtained sub-national estimates of the probability $_{10}q_5$. Working in a Bayesian inference setup allowed us to reflect and account for uncertainty in our sub-national estimates through their associated posterior distributions. The uncertainty correctly accounted for the complex survey design of DHS.

We obtained precise estimates for about two-thirds of period-specific sub-national units, defined as a CV below 20%. The sample size seemed to have the greater influence on the capacity of obtaining robust sub-national estimates than the number of surveys. Having these two criteria fulfilled, we observed that the age-specific risks of dying were highly and positively correlated. However, stratifying by period, in most cases, the Pearson correlations coefficients were not statistically significantly different from zero. Thus, for a given period, the Admin 1 entities with high $_5q_0$ were not necessarily those with high $_{10}q_5$. Hence, at the country level, the long term improvement in mortality drives the positive relationship between $_5q_0$ and $_{10}q_5$ but

**Table 1. Share (%) of total variance from model components.**

| Country | RW2 ($\sigma_\gamma^2$) | | ICAR ($\sigma_\phi^2$) | | RW2xICAR ($\sigma_\delta^2$) | | Time ($\sigma_\alpha^2$) | | Space ($\sigma_\theta^2$) | |
|---|---|---|---|---|---|---|---|---|---|---|
| | $_5\hat{q}_0$ | $_{10}\hat{q}_5$ | $_5\hat{q}_0$ | $_{10}\hat{q}_5$ | $_5\hat{q}_0$ | $_{10}\hat{q}_5$ | $_5\hat{q}_0$ | $_{10}\hat{q}_5$ | $_5\hat{q}_0$ | $_{10}\hat{q}_5$ |
| Benin | 81.7 | 84.4 | 9.8 | 1.7 | 3.2 | 10.3 | 3.1 | 1.9 | 2.2 | 1.7 |
| Burkina Faso | 47.5 | 18.7 | 28.0 | 9.0 | 17.0 | 63.1 | 5.3 | 5.3 | 2.2 | 3.8 |
| Cameroon | 37.6 | 33.6 | 56.0 | 47.0 | 3.2 | 16.5 | 2.6 | 2.2 | 0.7 | 0.8 |
| Ethiopia | 72.6 | 73.7 | 21.7 | 4.9 | 3.7 | 18.7 | 1.6 | 1.1 | 0.4 | 1.6 |
| Ghana | 56.0 | 36.1 | 34.4 | 3.0 | 6.4 | 53.0 | 2.7 | 5.1 | 0.5 | 2.7 |
| Guinea | 52.7 | 75.4 | 38.7 | 2.4 | 4.8 | 16.7 | 3.2 | 2.7 | 0.6 | 2.8 |
| Kenya | 30.3 | 20.7 | 50.5 | 29.1 | 15.9 | 38.0 | 2.2 | 2.4 | 1.1 | 9.8 |
| Lesotho | 43.0 | 9.6 | 6.8 | 1.9 | 33.8 | 84.6 | 8.5 | 1.9 | 8.0 | 2.0 |
| Madagascar | 76.4 | 67.6 | 2.4 | 3.8 | 5.6 | 20.6 | 14.6 | 5.4 | 1.0 | 2.5 |
| Malawi | 92.5 | 83.6 | 2.0 | 4.5 | 0.6 | 4.7 | 4.3 | 5.3 | 0.5 | 1.9 |
| Mali | 68.4 | 48.3 | 27.7 | 21.2 | 2.1 | 22.4 | 1.1 | 5.5 | 0.7 | 2.7 |
| Namibia | 16.9 | 4.1 | 38.1 | 24.6 | 35.3 | 68.3 | 4.5 | 2.3 | 5.2 | 0.8 |
| Niger | 54.6 | 3.8 | 36.3 | 6.8 | 7.0 | 81.2 | 1.3 | 5.3 | 0.8 | 2.8 |
| Nigeria | 33.9 | 17.3 | 60.1 | 46.5 | 3.5 | 24.7 | 1.6 | 1.7 | 0.9 | 9.7 |
| Rwanda | 89.9 | 92.2 | 1.2 | 1.2 | 0.8 | 2.7 | 7.5 | 3.2 | 0.6 | 0.7 |
| Senegal | 69.9 | 66.8 | 24.8 | 2.8 | 3.7 | 20.8 | 0.9 | 5.1 | 0.7 | 4.4 |
| Tanzania | 66.1 | 62.9 | 27.8 | 11.2 | 3.4 | 17.6 | 2.1 | 5.2 | 0.7 | 3.1 |
| Uganda | 88.6 | 63.0 | 2.5 | 2.7 | 2.6 | 23.3 | 5.7 | 2.3 | 0.6 | 8.7 |
| Zambia | 82.7 | 53.3 | 12.4 | 16.6 | 2.9 | 24.8 | 1.2 | 2.2 | 0.8 | 3.1 |
| Zimbabwe | 16.1 | 13.5 | 57.0 | 10.2 | 14.1 | 56.0 | 5.4 | 3.7 | 7.4 | 16.6 |

it is currently not possible to conclude that at the sub-national level, this relationship holds, expect in Cameroon, Ethiopia, Nigeria, Senegal and Zambia.

When decomposing the total variance associated with $logit(_5q_0)$ and $logit(_{10}q_5)$, we showed that the two age groups have a similar share of variation coming from the time trend (approximately half of the total variability). In contrast, the proportion of variation coming from the structured space term is approximately two times higher in $_5q_0$ compared to $_{10}q_5$. This reflects that sub-national variation within countries over all periods is higher for the younger age group. Burkina Faso, Ghana, Lesotho, Namibia, Niger and Zimbabwe are countries where more than 50% of their total variance is explained by the space×time interaction term. These countries also have a high share of sub-national estimates that are considered as imprecise. This suggests that in these cases, the method used does not allow to disentangle the signal from the noise. Hence, sub-national estimates for these countries have to be interpreted with caution.

This research is subject to some limitations. First, while we adjusted for bias related to the vertical transmission of HIV in under-five mortality rates, we did not account for such bias in older children and young adolescents. This bias should be smaller in older children than in children under age 5, as most children vertically infected will die before age 5 in the absence of treatment. Second, we worked with a somewhat arbitrary threshold for precision. To evaluate how the number of retained periods changed according to a more conservative threshold, we performed a sensitive analysis. Obviously, a more stringent criterion results in retaining fewer periods (91 periods with 20%, reduced to 50 periods with 15%). However, since the 95% CI around Pearson correlation coefficients accounted for uncertainty in sub-national estimates, we present uncertainty around correlation measures and the only change is the number of periods used, not the correlation estimate for a given period. Third, this study relied on a

limited number of countries as 20 countries fulfilled our selection criteria (i.e having at least 3 DHS representative at the sub-national level). Finally, we do not provide estimates at a finer geographical scale than the Admin 1 level. It is clear from our results that $_{10}q_5$ estimates at a finer level would be highly uncertain. There is a trade-off between geographical scale and precision of estimates [28].

In order to improve the measurement of mortality in older children at the sub-national level, other types of data sources are required, such as Health and Demographic Surveillance Systems (HDSS) (to study the relationship between $_5q_0$ and $_{10}q_5$ in detail), or reports on recent household deaths in censuses and summary birth history data (to reduce the uncertainty around space-time estimates). However, the methodology used here would need to be adapted as it was developed to be used on full birth histories. Another possibility that seems promising is to revise the approach and measure the $_{10}q_5$-to-$_5q_0$ ratio at the sub-national level, and model this ratio, instead of modelling the age-specific estimates as if they were independent and analyzing their relationship in a second step. The ratio of $_5q_0$ to $_{10}q_5$ could be modelled as a function of $_{15}q_0$ to inform estimates at the sub-national level in a similar way as Alexander and Alkema [33] model the ratio of neonatal mortality over post-neonatal mortality.

To conclude, our study assessed the feasibility of obtaining sub-national mortality estimates for children aged 5-14 with DHS surveys and a recent space-time smoothing method. We showed that sub-national estimates for this age group are surrounded by considerable uncertainty, even when combining surveys within countries. In Cameroon, Ethiopia, Nigeria, Senegal and Zambia, our study suggests that the spatial variations in mortality found in the under-5 age group mirror those in the 5-14 age group. A more extensive use of census data should help in establishing whether this is also the case in other countries.

## Supporting information

**S1 Appendix.**
(PDF)

## Acknowledgments

We thank Ashira Menashe-Oren and two anonymous reviewers for their valuable comments during the preparation of this manuscript.

## Author Contributions

**Conceptualization:** Benjamin-Samuel Schlüter, Bruno Masquelier.

**Data curation:** Benjamin-Samuel Schlüter.

**Formal analysis:** Benjamin-Samuel Schlüter.

**Funding acquisition:** Bruno Masquelier.

**Investigation:** Benjamin-Samuel Schlüter.

**Methodology:** Benjamin-Samuel Schlüter.

**Project administration:** Benjamin-Samuel Schlüter, Bruno Masquelier.

**Resources:** Benjamin-Samuel Schlüter, Bruno Masquelier.

**Software:** Benjamin-Samuel Schlüter.

**Supervision:** Bruno Masquelier.

**Validation:** Benjamin-Samuel Schlüter.

**Visualization:** Benjamin-Samuel Schlüter.

**Writing – original draft:** Benjamin-Samuel Schlüter, Bruno Masquelier.

**Writing – review & editing:** Benjamin-Samuel Schlüter, Bruno Masquelier.

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
