## [Decision Letter · Decision Letter 0]

30 Nov 2020

PONE-D-20-34856

Space-time smoothing of mortality estimates in children aged 5-14 in Sub-Saharan Africa

PLOS ONE

Dear Dr. Schlüter,

Thank you for submitting your manuscript to PLOS ONE. After careful consideration, we feel that it has merit but does not fully meet PLOS ONE’s publication criteria as it currently stands. Therefore, we invite you to submit a revised version of the manuscript that addresses the points raised during the review process.

Note that reviewer 1 recommendation to reject seems to stem basically from lack of perceived novelty and the analysis replicating methods from previous studies. Novelty and the perceived contribution of the field are not criteria for publication in PLOS ONE. Replication studies, on the other hand, are only considered to the extent that this replication is clearly acknowledged and a rationale is provided. The exploratory analysis of whether the methods can be applied to a related dimension of mortality is a possible rationale and the relationship of the manuscript to previous work is clearly acknowledged.

On the other hand, a main requisite is that the analysis is carried out to a high technical standard and is described in sufficient detail. Both reviews suggest changes to the manuscript in this spirit and raise concerns regarding the lack of detail in some aspects. Those should be addressed in the revision. Also that the conclusion drawn stem from the analysis and there are suggestions from the reviewers also in this respect.

In particular justification for the subnational classification and the countries chosen, or consideration of possible extension of the analysis as suggested by reviewer 1, and better documentation of methods to ensure replicability.

Regarding the countries chosen, you are providing a reason for your choice based on the number of DHS surveys. Since you are dealing with a subsample of countries from the studies you are replicating, it would be good to include an assesment of selectivity according to the main variables of study, i.e: 5q0. A table or a graph including mean levels in the last survey for the selected countries and the whole set of sub-Saharan African DHS countries or something of the sort might provide context to the replication.

We look forward to receiving your revised manuscript.

Kind regards,

José Antonio Ortega, Ph.D.

Academic Editor

PLOS ONE

Journal Requirements:

Reviewers' comments:

Reviewer's Responses to Questions

**Comments to the Author**

1. Is the manuscript technically sound, and do the data support the conclusions?

Reviewer #1: Partly

Reviewer #2: Yes

2. Has the statistical analysis been performed appropriately and rigorously? 

Reviewer #1: Yes

Reviewer #2: Yes

3. Have the authors made all data underlying the findings in their manuscript fully available?

Reviewer #1: Yes

Reviewer #2: Yes

4. Is the manuscript presented in an intelligible fashion and written in standard English?

Reviewer #1: Yes

Reviewer #2: Yes

5. Review Comments to the Author

Reviewer #1: This manuscript presents estimates of all-cause both-sex mortality children under five and children ages 5 through 15 from 10 sub-Saharan countries. The estimates were obtained using previously published modeling technology using publicly available data from DHS surveys. This manuscript is a partial repetition of the work reported in PONE 14(1):e210645, with the following differences:

1. The previously published work reported only under-5 mortality.

2. The ten countries for which mortality was estimated in the current manuscript are a subset of the 35 countries included in the previous publication.

3. The subnational estimates reported in the current manuscript are for arbitrarily defined "regions" rather than official geopolitical definitions (for examples, states for Nigeria and counties for Kenya).

The work reported in the current manuscript seems to lack novelty, other than provision of mortality for ages 5-14, and the utility of those estimates is compromised by the choice of arbitrary subnational units. The latter is unfortunate because rigorously produced estimates at the first and second subnational levels would be used to support ongoing public health efforts in sub-Saharan Africa.

I encourage the authors to pursue an expanded manuscript which would include the following features:

1. Use all years of DHS data for each country.

2. Produce subnational estimates for official geopolitical units. The minumum would be the first subnational units, which are targets of estimation for the DHS surveys. Increasingly, health interventions are being planned and implemented at the second subnational level, so that level should be a viable target for estimation.

3. Consider comparing the national-level model based estimates with census-based estimates (i.e., UN IGME estimates).

Minor points:

1. I did not find any actual modelling code in the GitHub repository. Authors do the scientific community and their own credibility a great service by providing complete code along with some sort of description of the sequence of execution.

2. Lines 117-118 beg for a full mathematical description of the logistic model, perhaps as supporting information.

Disclaimer: All reviewers and reviews are prone to error and misunderstanding.

Reviewer #2: The manuscript conducted analysis of subnational estimates of child mortality (5q0 and 10q5) using 37 DHS surveys in 10 Sub-Saharan countries. A space-time smoothing model that has been previously used for 5q0 is estimated here. The main modeling part is well implemented. The data, method, and study designs have been clearly described as well. In general I think this is a well-written paper that extended previous work. I think there are a few places where more careful interpretation and discussions are needed. My major comments are as follows:

1. The use of 'robustness' when describing the estimates could be confusing to some audience. In statistics, robustness refers to the insensitivity of the estimators to model assumptions. From what I understand, the procedure of calculating CV is to assess whether the estimates have too much uncertainty to be useful. Such posterior uncertainties are directly translated from the design-based uncertainty, which is most directly related to sample size. So when you say an estimate is not robust, I think it means it cannot be estimated with enough precision given the data. Do you agree with the interpretation? Since it has nothing to do with the property of the estimators under model misspecification, I think terming that as 'robustness' is inappropriate.

2. Related to the comment above, as you mentioned in Line 333 and when interpreting the results, data sparsity potentially makes estimating 10q5 more difficult than 5q0. Could you give more information how that affects your interpretation of the results? For example, I would expect the direct estimates to have high variance in general. I think it might be useful to visualize the direct estimates and their uncertainty as well in the supplement, so that readers can have a sense of how much uncertainty comes from the first stage estimates, and how much of a reduction the smoothing model led to.

3. Except for Nigeria, the spatial component in the model explained much less variations in 10q5 compared to 5q0. I agree with the interpretation in the manuscript that this is likely due to higher levels of noises in the data. I think this leads to the question how much to trust the trends and spatial heterogeneity of the estimates, i.e., whether the space-time interactions are guided by true subnational differences, or are they more influenced by noise. Without ground truth I understand this is not a question that can be definitively answered, but I think a more clear message is useful for readers to know this caveat. Again plotting the raw estimates may also help here.

4. You may consider adding a section in the supplement that gives more details of the model, e.g., prior specifications etc. While I expect such things can be inferred from the replication codes by informative readers who is familiar with R, it is useful to put them in writing for the audience of the paper who are not from a computational field.

Minor comment:

- Line 272, "confidence interval" should be "credible interval".

6. PLOS authors have the option to publish the peer review history of their article (what does this mean?). If published, this will include your full peer review and any attached files.

Reviewer #1: No

Reviewer #2: No

---

## [Author Response · Author response to Decision Letter 0]

22 Dec 2020

Dear José Antonio Ortega,

We are grateful that you offered us an opportunity to revise and resubmit our paper entitled Space-time smoothing of mortality estimates in children aged 5-14 in Sub-Saharan Africa for possible publication in PLOS ONE. We would like to express our sincere thanks to the two reviewers who pointed out areas that needed clarification in the manuscript. 

We believe that we have addressed all comments provided by the reviewers. You will find below a point-by-point response to their comments. The main changes to the article refer to the addition of 59 DHS surveys, reaching a total of 96 DHS surveys conducted in 20 countries. This has been allowed by a relaxation of our selection criteria to respond to a comment form the first reviewer. We have improved our wording with respect to precision assessment (instead of robustness assessment) and have avoided using the term “region” as we were working at the level of Admin 1. We also added more contextual information in the supporting information such as comparisons with UN IGME 10q5 estimates and a selectivity assessment of our sample in comparison to the one used by Li and colleagues (2019). Finally, in the discussion section, we now mention that the method does not seem to be able to provide reliable sub-national estimates for some countries, as suggested by the high amount of variance explained by the space-time interaction.

We hope that you will find the revised manuscript acceptable for publication in PLOS ONE, and we look forward to your appraisal.

Best regards, 

Benjamin Schluter

Reviewer #1: This manuscript presents estimates of all-cause both-sex mortality children under five and children ages 5 through 15 from 10 sub-Saharan countries. The estimates were obtained using previously published modeling technology using publicly available data from DHS surveys. This manuscript is a partial repetition of the work reported in PONE 14(1):e210645, with the following differences:

1. The previously published work reported only under-5 mortality.

2. The ten countries for which mortality was estimated in the current manuscript are a subset of the 35 countries included in the previous publication.

3. The subnational estimates reported in the current manuscript are for arbitrarily defined "regions" rather than official geopolitical definitions (for examples, states for Nigeria and counties for Kenya).

The work reported in the current manuscript seems to lack novelty, other than provision of mortality for ages 5-14, and the utility of those estimates is compromised by the choice of arbitrary subnational units. The latter is unfortunate because rigorously produced estimates at the first and second subnational levels would be used to support ongoing public health efforts in sub-Saharan Africa. 

I encourage the authors to pursue an expanded manuscript which would include the following features:

We thank the reviewer for this encouragement and have tried to address all her/his specific comments below.

1. Use all years of DHS data for each country.

Our paper has been substantially revised as a respond to this comment. In order to fulfill the reviewer’s request, we slightly changed our selection criteria for DHS surveys. We still selected a country if the country had conducted at least 3 DHS surveys with representative data at a sub-national level. This condition is necessary to pool information across sufficient DHS surveys to increase the amount of information used to model sub-national mortality rates for ages 5-14. However, we relaxed the condition that three successive surveys were needed to have a consistent set of sub-national boundaries. To do so, we used the GPS coordinates of each cluster to re-locate clusters in coherent Admin 1 units across surveys.

This allowed us to add DHS surveys for countries previously included but also to add additional countries in the analysis. Our sample increased from 10 to 20 countries, using now 96 DHS in total. Out of the 20 countries included in our analysis, 17 use the same sub-national units as in the study by Li and colleagues (2019) that we are replicating for older children and young adolescents. For the three countries that remain, we stayed at the Admin 1 level while Li and colleagues (2019) used Admin 2 level. In regards to our results, estimating at this level would be asking to much from the data. Our sample is still smaller than in the study by Li and colleagues (2019). This is because they included two countries in Northern Africa (Egypt and Morocco), while our study is focusing on Sub-Saharan Africa. Considering the differences in mortality levels across these two regions, and the fact that only two countries cover Norther Africa, we believe it is preferable to limit the analysis to Sub-Saharan Africa. In the 13 other countries included in their study but excluded from ours, there were less than 3 DHS surveys which were representative at a subnational level. Again, considering our results, including those data-sparse countries in our study would be asking too much of the data. 

2. Produce subnational estimates for official geopolitical units. The minimum would be the first subnational units, which are targets of estimation for the DHS surveys. Increasingly, health interventions are being planned and implemented at the second subnational level, so that level should be a viable target for estimation.

We thank the reviewer for highlighting a confusion we might have created in the manuscript. We are using the first sub-national units defined by DHS surveys which are Admin 1. However, we realized that we were using the term “region” in the manuscript to describe such sub-national units which might have created confusion. We now use “Admin 1” and “sub-national” units and avoided the term “region”. Note that for some countries, we are not using the Admin-1 division used by latest DHS survey but rather the administrative subdivision of the past. This comes from the fact that for a given country, we needed a coherence in our sub-national units over time in order to smooth and pool information during the estimation process. In most countries included in our study, changes over time have been modest. 

As for the Admin-2 level, our results suggests that another modelling approach would be required, moving from an area-level approach to a continuous spatial model, as we would have numerous units with very small counts of deaths and person-years. The use of covariates would perhaps also be required in this case. Our objective here was first to evaluate the reliability of estimates at the Admin-1 level. 

3. Consider comparing the national-level model based estimates with census-based estimates (i.e., UN IGME estimates).

We added a section in the appendix where we compared national-level model estimates with those estimated by UN IGME, based on the B3 model. The UN IGME estimates are based in part on the same DHS surveys, but are also informed by other data series, such as MICS surveys or reports from censuses on recent household deaths. In addition, the B3 model used for reconstructing trends in mortality includes a data model that captures recall and truncation biases. There is evidence that DHS surveys tend to slightly under-estimate mortality in the age group 5-14, when compared to other data series. As a result, the final estimates are adjusted upwards and may fall above the original survey data points. This explains why the UN IGME estimates tend to be higher than our estimates from the national-level model, based solely on DHS without adjustment for non-sampling errors. We refer the reader to that section in the manuscript (lines 214-217).

Minor points:

1. I did not find any actual modelling code in the GitHub repository. Authors do the scientific community and their own credibility a great service by providing complete code along with some sort of description of the sequence of execution.

We are sorry that the reviewer could not access our code. The repository was already created but we were waiting for reviewers’ comments and last modifications of the analysis to upload the final codes. The latest code is now uploaded and available for review. 

2. Lines 117-118 beg for a full mathematical description of the logistic model, perhaps as supporting information.

We provide additional mathematical details see eq (2), eq (3) and lines 160-162 in the main manuscript.

Reviewer #2: The manuscript conducted analysis of subnational estimates of child mortality (5q0 and 10q5) using 37 DHS surveys in 10 Sub-Saharan countries. A space-time smoothing model that has been previously used for 5q0 is estimated here. The main modeling part is well implemented. The data, method, and study designs have been clearly described as well. In general I think this is a well-written paper that extended previous work. 

We thank the reviewer for his/her appreciation of our work and respond to comments below.

I think there are a few places where more careful interpretation and discussions are needed. My major comments are as follows:

1. The use of 'robustness' when describing the estimates could be confusing to some audience. In statistics, robustness refers to the insensitivity of the estimators to model assumptions. From what I understand, the procedure of calculating CV is to assess whether the estimates have too much uncertainty to be useful. Such posterior uncertainties are directly translated from the design-based uncertainty, which is most directly related to sample size. So when you say an estimate is not robust, I think it means it cannot be estimated with enough precision given the data. Do you agree with the interpretation? Since it has nothing to do with the property of the estimators under model misspecification, I think terming that as 'robustness' is inappropriate.

We fully agree with the reviewer’s interpretation. We rephrased it as “precision” and changed the text accordingly.

2. Related to the comment above, as you mentioned in Line 333 and when interpreting the results, data sparsity potentially makes estimating 10q5 more difficult than 5q0. Could you give more information how that affects your interpretation of the results? For example, I would expect the direct estimates to have high variance in general. I think it might be useful to visualize the direct estimates and their uncertainty as well in the supplement, so that readers can have a sense of how much uncertainty comes from the first stage estimates, and how much of a reduction the smoothing model led to.

We added the comparison between the estimates obtained after the meta-analysis and the ones obtained after space-time smoothing in the supporting information (see section S3). Note that since we used multiple DHS surveys for a given country, the meta-analysis step already pooled some information together for a given Admin 1-period combination. We also compared the direct estimates for one survey in the introduction: “For example, in the 2018 DHS conducted in Zambia, the under-five mortality rate for the period 0-4 years before the survey was 58.1 per thousand at the national level, with a standard error of 3.6 per thousand (coefficient of variation = 6.3%). The corresponding risk of dying in the age group 5-14 was 10.0 per thousand, with a standard error of 1.4 (CV= 14.0%). For this recent period, women aged 15-49 in the survey reported more than 7 times more deaths among children aged 0-4 years than among children aged 5-14 years, for a relatively comparable exposure time (1.5 times more person-years in children aged 5-14).” It is interesting to note that the number of person-years is not twice higher in children aged 5-14, when compared to those aged 0-4. This is because of a truncation bias as only a fraction of women aged 15-49 will have children aged 5-14 to report on. 

3. Except for Nigeria, the spatial component in the model explained much less variations in 10q5 compared to 5q0. I agree with the interpretation in the manuscript that this is likely due to higher levels of noises in the data. I think this leads to the question how much to trust the trends and spatial heterogeneity of the estimates, i.e., whether the space-time interactions are guided by true subnational differences, or are they more influenced by noise. Without ground truth I understand this is not a question that can be definitively answered, but I think a more clear message is useful for readers to know this caveat. Again plotting the raw estimates may also help here.

We thank the reviewer for this comment. We expanded the discussion in order to provide a clear message (see lines 392-397). We do not see how plotting the estimates would help, as these will be much more irregular than those obtained by pooling surveys together and smoothing estimates across time and space. 

4. You may consider adding a section in the supplement that gives more details of the model, e.g., prior specifications etc. While I expect such things can be inferred from the replication codes by informative readers who is familiar with R, it is useful to put them in writing for the audience of the paper who are not from a computational field.

In line with the comment of reviewer 1, we expanded the mathematical details of the model (see eq (2) and eq (3)). Regarding the priors, the specification of the prior where in the manuscript but they were not clearly stated as such. We improve clarity and provided additional details (see lines 184-201). 

Minor comment:

- Line 272, "confidence interval" should be "credible interval".

This has been modified, thank you.

Editorial comment: Regarding the countries chosen, you are providing a reason for your choice based on the number of DHS surveys. Since you are dealing with a subsample of countries from the studies you are replicating, it would be good to include an assesment of selectivity according to the main variables of study, i.e: 5q0. A table or a graph including mean levels in the last survey for the selected countries and the whole set of sub-Saharan African DHS countries or something of the sort might provide context to the replication.

Thank you for this suggestion. Please see our response to the first reviewer about the expansion of our sample to 20 countries. We also added a graph in the supplementary material comparing the estimates of 10q5 in the countries in the sample of Li and colleagues (2019) and this study. In the methods section, we also added a note on how our sample compares with all countries in Sub-Saharan in terms of mortality: “The population-weighted average of the risk of dying in the age group 5-14 was 20 per 1000 in 2019 in these 20 countries, which is slightly higher than the regional estimate for Sub-Saharan Africa (16 per 1000) (UN IGME 2020). In 2019, 65% of all deaths that occurred in Sub-Saharan Africa in the age group 5-14 where in these 20 countries.”

---

## [Decision Letter · Decision Letter 1]

5 Jan 2021

Space-time smoothing of mortality estimates in children aged 5-14 in Sub-Saharan Africa

PONE-D-20-34856R1

Dear Dr. Schlüter,

We’re pleased to inform you that your manuscript has been judged scientifically suitable for publication and will be formally accepted for publication once it meets all outstanding technical requirements.

Kind regards,

José Antonio Ortega, Ph.D.

Academic Editor

PLOS ONE

Additional Editor Comments (optional):

Reviewers' comments:

Reviewer's Responses to Questions

**Comments to the Author**

1. If the authors have adequately addressed your comments raised in a previous round of review and you feel that this manuscript is now acceptable for publication, you may indicate that here to bypass the “Comments to the Author” section, enter your conflict of interest statement in the “Confidential to Editor” section, and submit your "Accept" recommendation.

Reviewer #1: All comments have been addressed

Reviewer #2: All comments have been addressed

2. Is the manuscript technically sound, and do the data support the conclusions?

Reviewer #1: Yes

Reviewer #2: Yes

3. Has the statistical analysis been performed appropriately and rigorously? 

Reviewer #1: Yes

Reviewer #2: Yes

4. Have the authors made all data underlying the findings in their manuscript fully available?

Reviewer #1: Yes

Reviewer #2: Yes

5. Is the manuscript presented in an intelligible fashion and written in standard English?

Reviewer #1: Yes

Reviewer #2: Yes

6. Review Comments to the Author

Reviewer #1: Well done! (These are more characters to satisfy the minimum character count: kgjfdlghfioo;ajkklfj;)

Reviewer #2: The authors have addressed all my comments. I have nothing further to add. I congratulate the authors on conducting this analysis.

7. PLOS authors have the option to publish the peer review history of their article (what does this mean?). If published, this will include your full peer review and any attached files.

Reviewer #1: No

Reviewer #2: No

---

## [Editor Report · Acceptance letter]

7 Jan 2021

PONE-D-20-34856R1 

Space-time smoothing of mortality estimates in children aged 5-14 in Sub-Saharan Africa 

Dear Dr. Schlüter:

I'm pleased to inform you that your manuscript has been deemed suitable for publication in PLOS ONE. Congratulations! Your manuscript is now with our production department. 

Kind regards, 

on behalf of

Dr. José Antonio Ortega 

Academic Editor

PLOS ONE